# Impact of Char Properties and Reaction Parameters on Naphthalene Conversion in a Macro-TGA Fixed Char Bed Reactor

**Ziad Abu El-Rub** [1,*], **Eddy Bramer** [2], **Samer Al-Gharabli** [1] and **Gerrit Brem** [2]

1   Pharmaceutical and Chemical Engineering Department, German Jordanian University, Amman 11180, Jordan; samer.gharabli@gju.edu.jo

2   Laboratory of Thermal Engineering, University of Twente, P. O. Box 217, 7500 AE Enschede, The Netherlands; e.a.bramer@utwente.nl (E.B.); g.brem@utwente.nl (G.B.)

*   Correspondence: ziad.abuelrub@gju.edu.jo; Tel.: +962-6-429-4412

**Abstract:** Catalytic tar removal is one of the main challenges restricting the successful commercialization of biomass gasification. Hot gas cleaning using a heterogeneous catalyst is one of the methods used to remove tar. In order to economically remove tar, an efficient low-cost catalyst should be applied. Biomass char has the potential to be such a catalyst. In this work, the reactor parameters that affect the conversion of a model tar component "naphthalene" were investigated employing an in situ thermogravimetric analysis of a fixed bed of biomass char. The following reactor and catalyst parameters were investigated: bed temperature (750 to 900 °C), gas residence time in the char bed (0.4 to 2.4 s), char particle size (500 to 1700 μm), feed naphthalene concentration, feed gas composition (CO, $CO_2$, $H_2O$, $H_2$, $CH_4$, naphthalene, and $N_2$), char properties, and char precursor. It was found that the biomass char has a high activity for naphthalene conversion. However, the catalytic performance of the biomass char was affected by the gasification reactions that consumed its carbon, and the coke deposition that reduced its activity. Furthermore, high ash and iron contents enhanced char activity. The results of this work will be used in the design of a process that uses biomass char as an auto-generated catalyst in the gasification process.

**Keywords:** biomass char; catalyst; naphthalene; fixed bed reactor; macro-TGA; gasification

## 1. Introduction

The world's dependence on fossil fuels, including crude oil, natural gas, and coal, reached 81.4% of total primary energy sources compared to 9.7% of biofuels and waste in the year 2015 [1]. The high consumption of fossil fuels has led to a sharp decrease in global reserves, severe fluctuation in oil prices, as well as a negative environmental impact. Therefore, there is an immense need to increase the share of renewable and environmentally friendly sources of energy that reduce the drawbacks of fossil fuels.

Biomass is a generic term for organic materials often derived from living or recently dead plants and animals. It is considered to be a promising renewable energy source, similar to solar, wind, geothermal, and hydropower sources [2]. Biomass energy has received substantial attention due to its abundance and its potential to reduce global greenhouse emissions [3]. Several chemical-thermal technologies can be used to exploit the energy stored in biomass. Gasification is one notable technology, where biomass is converted by partial combustion into a combustible gas. However, this technology still faces technical challenges that delay its commercialization. The removal of tar generated in the gasifier is considered the most challenging obstacle due to the operational problems it causes downstream of the gasifier.

Tar is a generic name that includes all organic compounds present in the producer gas with a molecular weight higher than benzene [4]. Their amount and composition in the producer gas depend on the gasification conditions and the type of biomass. The tar can be removed by physical, thermal cracking, and catalytic conversion processes. The last method, where a catalyst is applied under a temperature close to that of the gasifier, is considered the most economical one [5].

The suitability of a catalyst for the commercial application of tar conversion depends on its activity, selectivity, stability, mechanical strength, and cost [6]. Biomass char is a byproduct of the biomass gasification process. It shows an effective catalytic activity for tar removal that can be related to its surface characteristics (e.g., surface area and pore size) and mineral content. Thus, this auto-generated material from the gasification process can be used as a primary or secondary catalytic measure. However, a comprehensive understanding of its catalytic activity and impact of reaction conditions are needed in order to design a reliable char-based process for hot gas cleaning. Sudarsanam et al. [7] presented a comprehensive review of recent advances in catalytic biomass conversions in addition to applications of a biomass-derived catalyst including biomass char. Moreover, Park et al. [8] found that using biomass char in the gasifier (as a primary measure) cannot eliminate the need for a subsequent unit downstream of the gasifier for removing the tar completely (as a secondary measure).

Fuentes-Cano et al. [9] found that the char catalytic activity is strongly dependent on the concentration of alkali metals (e.g., K and Na) and alkaline earth metals (e.g., Ca). Furthermore, they reported that the absence of these metals caused a quick char deactivation. However, Nzihou et al. [10] related the catalytic influence of the alkali and alkaline metals to their release at elevated temperatures and the subsequent dispersion of these metals in the carbon matrix.

Greensfelder et al. [11] reported that the biomass char converts the tar into free radical by removing the hydrogen. The active free radicals undergo heavy polymerization reactions that produce coke deposited on the char surface [12]. In general, the severity of coke formation can be related to the number of rings in the tar component as stated by Hosokai et. al. [12]. However, Tenser et al. found that naphthalene is, specifically, the highest in terms of coke formation. Furthermore, Nestler et al. [12] tested the decomposition of naphthalene over different wood char. They found that naphthalene was completely converted to carbon and hydrogen, as they did not detect other gaseous components except hydrogen using gas chromatography (GC).

Figure 1 shows a scheme of the homogenous and heterogeneous reactions that occur on the surface of the char particle and the atmosphere of its producer gas. There is one main homogenous gas phase reaction, which is the water-gas shift reaction. Whereas, the biomass char particle undergoes several heterogeneous gasification reactions, which are mainly the water-gas reaction $(C + H_2O \rightarrow CO + H_2)$ and the Boudouard reaction $(C + CO_2 \rightarrow 2CO)$. On one hand, the tar is subjected to other heterogeneous reactions, which are mainly the steam reforming (tar + steam) and dry reforming (tar + $CO_2$) reactions. These reactions are activated by the active sites on the char surface to produce CO and $H_2$. However, they might also produce reactive coke that is consumed by these reforming reactions [13]. On the other hand, tar can be adsorbed on the char surface to form free radicals by removing the hydrogen. These free radicals can enter heavy polymerization reactions to produce less reactive coke, which deactivates the char surface active sites [13].

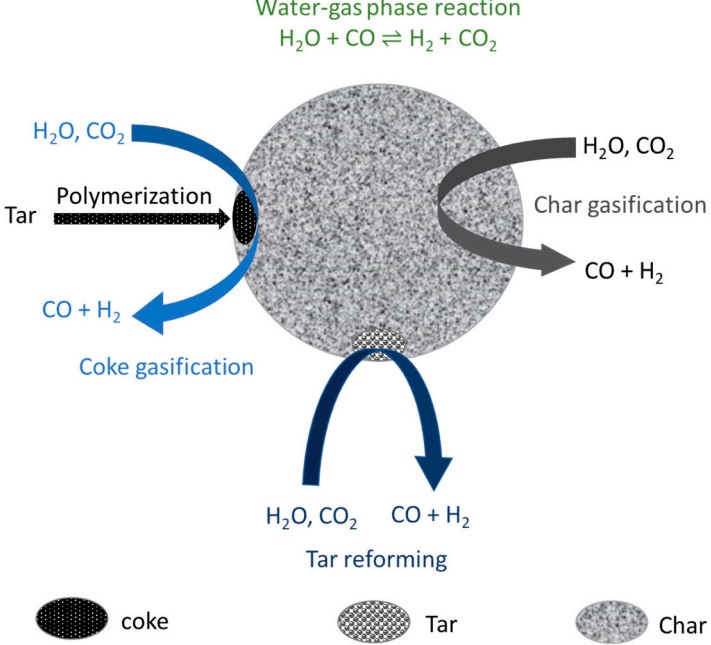

**Figure 1.** Scheme of the homogenous reaction (water-gas shift reaction) and qualitative (unbalanced) heterogeneous reactions (tar polymerization, tar reforming, char gasification, coke gasification) on the surface and atmosphere of the char particle.

In a previous study [14], a single particle model of biomass char was developed. This model studied the activity of the biomass char particle, naphthalene conversion, and the concurrent carbon conversion of the particle. Table 1 summarizes the main reactions considered. It was found that there was no significant effect for internal or external mass transfer limitations on the naphthalene and carbon conversion. However, the conversion reactions affected the physical properties and pore structure of the char particle. Furthermore, the char particle could be assumed isothermal up to a bulk temperature of 1160 °C.

**Table 1.** Main reactions occurring in a fixed char bed reactor for treating naphthalene as a model tar component.

| | Reaction | Rate of Reaction ($kmol \cdot m^{-3} \cdot s^{-1}$) | Reference |
|---|---|---|---|
| 1. | $H_2O + CO \rightleftharpoons H_2 + CO_2$ | $-r_1 = 2.78 \cdot 10^3 e^{-\frac{12,560}{RT}} C_{H_2O} \cdot C_{CO} - 1.05 \cdot 10^5 e^{-\frac{45,466}{RT}} C_{H_2} \cdot C_{CO_2}$ | [15] |
| 2. | $C_{10}H_8 + 10H_2O \rightarrow 14H_2 + 10CO$ | $-r_2 = \eta \cdot 10^{-4} e^{-\frac{61,000}{RT}} \cdot C_{C_{10}H_8} \cdot a$ | [16] |
| 3. | $C + H_2O \rightarrow CO + H_2$ | $-r_3 = 5.09 \cdot 10^4 e^{-\frac{238,000}{RT}} C_{H_2O} \cdot C_C$ | [17] |
| 4. | $C + CO_2 \rightarrow 2CO$ | $-r_4 = 1.12 \cdot 10^8 e^{-\frac{245,000}{RT}} C_{CO_2} \cdot C_C$ | [18] |
| 5. | $C + 2H_2 \rightarrow CH_4$ | $-r_5 = 9.14 \cdot 10^{-10} e^{-\frac{149,050}{RT}} C_{CO_2}$ | [19] |

a: char catalyst activity; η: effectiveness factor; R: gas constant; T: temperature.

The objective of this work is to study the impact of secondary measures on naphthalene and char conversion in a macro thermal gravimetric analysis (macro-TGA) fixed char bed reactor. These measures include the impact of reaction parameters and char properties. The studied parameters were char bed temperature, gas residence time in the char bed, char particle size, feed naphthalene concentration, feed gas composition, char surface area, and char precursor. The outcome of this research will help in the design of an integrated process that utilizes the byproduct biomass char as a catalyst in the gasification process.

## 2. Results and Discussion

### 2.1. Impact of Temperature

Figure 2 shows the impact of char bed temperature on naphthalene conversion. The naphthalene conversion at 15 min operating time was in the range of 70 wt.% and 100 wt.% and in the temperature range of 700 and 900 °C, respectively. This is comparable with results of Fuentes-Cano et al. [9] who achieved almost 100 wt.% naphthalene conversion at 900 °C at 0.15 s residence time for activated coal char with steam for 150 min. Whereas, Park et al. [8] found 75 wt.% maximum primary tar conversion in the gasifier at 800 °C using hot char particles. In addition, Nestler et al. [12] found also lower naphthalene conversion (76 wt.%) at 850 °C for wood char pyrolyzed at 500 and 800 °C.

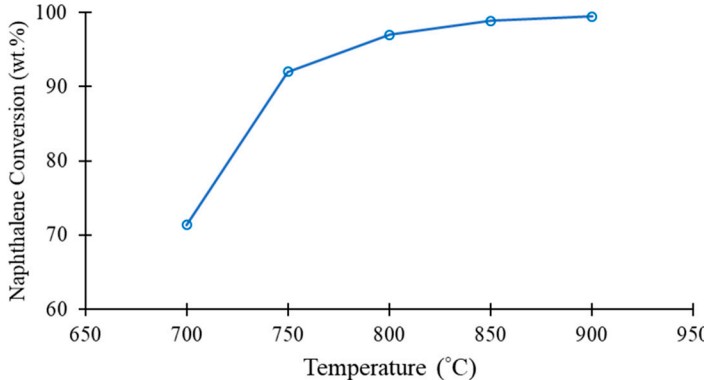

**Figure 2.** Impact of char bed temperature on naphthalene conversion. $\tau$ = 0.3 s; $d_p$ = 500–800 μm, t = 15 min.

Figure 3 shows the impact of reactor temperatures of 750 °C and 900 °C on the conversion of naphthalene during the eight hours experimental operating time. The naphthalene conversion at 900 °C remained at almost 100 wt.%, which can be related to the high activity of char at this temperature and the continuous activation of char with steam and $CO_2$.

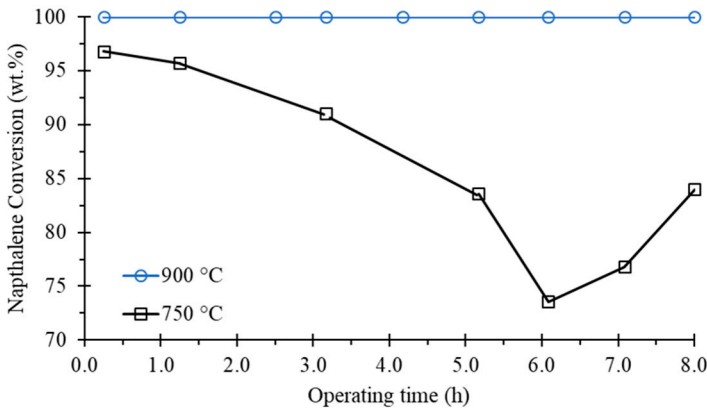

**Figure 3.** Impact of reactor temperature on naphthalene conversion during the operating time. $\tau$ =1.2 s; $d_p$ = 500–630 μm.

On the other hand, the naphthalene conversion at 750 °C decreased with time from 96 wt.% to 74 wt.% at the sixth hour then increased to 84 wt.% at the eighth hour. This can be related to the lower char activity at this temperature resulting from the coke formation, which reduces access to the active surface area of the char. The catalytic activity of the char particles is affected on one hand by the char activation, which resulted in carbon loss because of its reaction with steam and carbon dioxide. This activation exposes more active sites and surface metals that promote reforming reactions

of naphthalene. On the other hand, the char is deactivated by the coke formation resulting from the naphthalene cracking reactions. Therefore, the later increase in naphthalene conversion after the sixth hour might be related to the positive net effect.

The concurrent carbon conversion at the bed temperature range of 700–850 °C is shown in Figure 4. The increased carbon conversion from 2 wt.% at 800 °C to 7 wt.% at 850 °C indicates that the bed temperature is a key parameter, which is an indication that the reaction is kinetically limited.

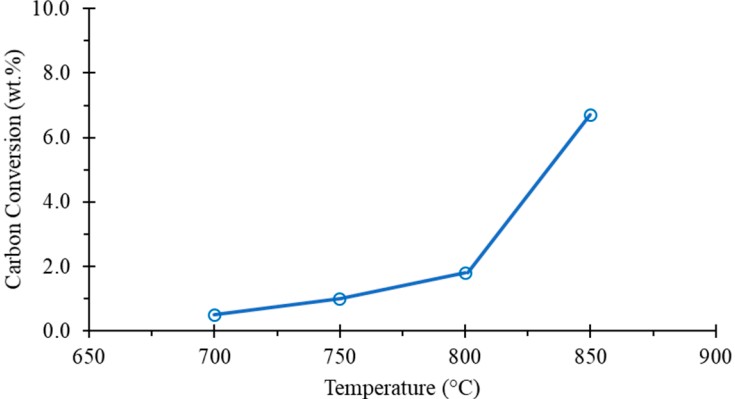

**Figure 4.** Impact of bed temperature on carbon conversion. $\tau$ = 1.2 s; $d_P$ = 500–630 μm; t = 1 h.

Figure 5 shows the impact of reactor temperature on carbon conversion during the eight-hour operating time at different gas residence times. At 750 °C and 1.2 s residence time, there was a negligible measurable loss in carbon conversion. This can be related to either a low gasification rate at this temperature or a negligible net weight of the gasified carbon in char and the produced carbon in the coke resulting from naphthalene cracking reactions.

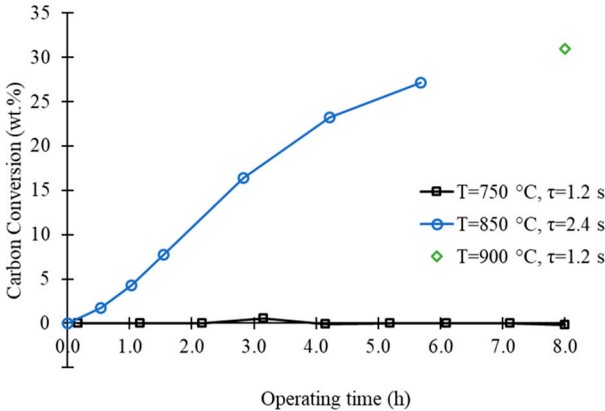

**Figure 5.** Impact of reactor temperature on carbon conversion during the operating time at different gas residence times. $d_P$ = 500–630 μm.

At 850 °C and a longer gas residence time (2.4 s), the carbon conversion increased to 27 wt.% after 5.7 h operating time. However, it was better to vary one parameter only (temperature) and fix the value of gas residence time because both of them have a positive influence on carbon conversion.

At 750 °C and 1.2 s residence time, the weight of carbon was measured only at the end of the eight-hour experiment. The weight loss was found to be 31 wt.%. It was not much higher than that of the 850 °C, due to the lower gas residence time here.

## 2.2. Impact of Gas Residence Time

The conversion of naphthalene can vary with the gas residence time as naphthalene conversion reactions can be treated as a first-order reaction. The residence time, or more precisely the space-time, can be experimentally manipulated by either changing the height of the char bed or the velocity of the feed gas. Figure 6 shows the impact of the gas residence time on the conversion of naphthalene at a relatively low reactor temperature of 750 °C. The small increase in the naphthalene conversion can be related to the narrow range of the studied gas residence time.

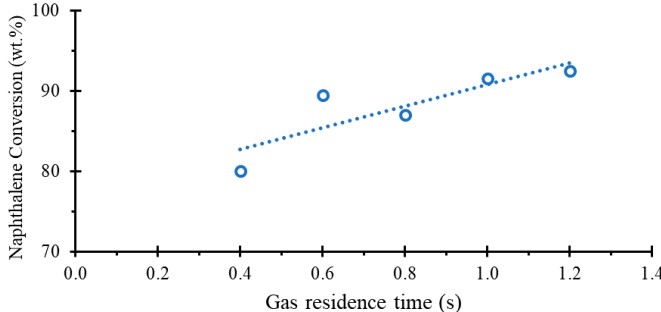

**Figure 6.** Impact of gas residence time on naphthalene conversion. T = 750 °C; $d_p$ = 1180 μm; t = 15 min.

## 2.3. Impact of Feed Naphthalene Concentration

Naphthalene conversion reactions include cracking reactions that produce coke, which deactivates the activity of the char. Therefore, the impact of feed naphthalene loading on its conversion was tested. The test was conducted at 6 to 18 $g \cdot Nm^{-3}$ feed naphthalene concentration, 0.15 s gas residence time, 900 °C, 1400–1700 μm char particle size, and 15 min operating time. There was no significant effect of the feed naphthalene concentration as the conversion was almost constant (89 wt.%–91 wt.%). This showed that the naphthalene-char reaction was first order because: (i) it was independent of the initial naphthalene concentration in these process conditions, (ii) no significant char consumption occurred during the studied operating time, and (iii) most heterogeneous gas phase reactions are first order. However, for longer operating times, the deposition of coke resulting from naphthalene cracking reactions can cause a serious decline in char activity.

As coke can be formed by naphthalene cracking reactions, it becomes important to confirm its formation and its negative effect on char gasification. Accordingly, two one-hour experiments were conducted at 850 °C, with conditions of 0.3 s gas residence time and 565 μm average particle size for 60 min. operating time and a feed gas mixture composition is given in Table 2. The first experiment was conducted in the absence of naphthalene and resulted in 65 wt.% carbon conversion. Whereas, the second experiment was conducted with naphthalene and resulted in 30 wt.% carbon conversion, which was less than half of the first experiment. This means that naphthalene in the second experiment was either (1) converted to coke (mainly carbon) that reduced the net carbon loss, or (2) the naphthalene cracking reactions formed less reactive coke that deactivated the char and thus reduced the rate of its gasification reactions. To verify this interpretation, it was assumed that all the carbon in the naphthalene was converted into coke. The first explanation is achieved if the converted naphthalene to coke matches the difference in the carbon weight between the two experiments. The difference in measured carbon weight was found to be 1.3 g, whereas the amount of carbon in the fed naphthalene was estimated to be 0.9 g. Thus, there was more carbon (0.4 g) than fed carbon in the naphthalene, which was not gasified. As the first explanation was proven to be not possible, the second explanation can be considered the correct one.



**Table 2.** Test conditions in the macro-TGA fixed char bed.

|  |  | Symbol | Value |
|---|---|---|---|
| | Pressure | $P$ (Pa) | $1.01 \times 10^5$ |
| | Bed Temperature | $T$ (°C) | 700–900 |
| Operating parameter | Gas space-time | $\tau$ (s) | 0.4–1.2 |
| | Average char particle size | $d_p$ (μm) | 565–1200 |
| | Operating time | $t$ (min) | 15–480 |
| | Volume fraction | | |
| | CO | - | 6.0% |
| | $CO_2$ | - | 10.0% |
| | $H_2O$ | - | 7.0% |
| Feed gas composition | $H_2$ | - | 4.0% |
| | $CH_4$ | - | 2.4% |
| | Naphthalene | - | 10–20 g/Nm$^3$ |
| | $N_2$ | - | Balance |

*2.4. Impact of Gas Composition*

The gas composition of the producer gas can affect both the tar (naphthalene) and char conversions. Naphthalene can be converted by two main reforming reactions: (1) steam reforming $(C_{10}H_8 + 10H_2O \rightarrow 14H_2 + 10CO)$, and (2) dry reforming $(C_{10}H_8 + 10CO_2 \rightarrow 20CO + 4H_2)$. The impact of the gas composition on the naphthalene conversion at 750 °C is shown in Table 3. As steam and carbon dioxide are the main reactive components with naphthalene, each component was studied separately at 10 vol.%, which is a common concentration in the producer gas. It was found that the dry reforming reaction gave the highest naphthalene conversion (97 wt.%). However, the other gas combinations gave the same naphthalene conversion (93 wt.%). This could mean that the carbon dioxide made the char surface more active towards naphthalene conversion reactions by creating more active sites than in cases of other gas combinations. Furthermore, it was reported that the presence of other components ($H_2$, CO, $CH_4$) can cause an inhibitory effect [20,21].

**Table 3.** Impact of the gas composition on naphthalene conversion. T = 750 °C; gas balance is $N_2$.

| Gas Composition (Volume Fraction) | | | | | Naphthalene Mass Conversion |
|---|---|---|---|---|---|
| $H_2O$ | $CO_2$ | $H_2$ | CO | $CH_4$ | |
| 10% | - | - | - | - | 93% |
| - | 10% | - | - | - | 97% |
| 10% | 10% | - | - | - | 93% |
| 7 % | 10% | 4% | 6% | 2.45 | 93% |

The impact of gas composition on carbon conversion during a 90 min operating time is given in Figure 7. Although $CO_2$–$N_2$ gas gave the highest naphthalene conversion, here $H_2O$–$N_2$ gas gave the maximum carbon conversion. The rest of the gas compositions gave almost comparable char conversions. This can be related to the ability of the water-gas reaction $(C + H_2O \rightarrow CO + H_2)$ to make the char more reactive than in other char-gasification reactions (4 and 5) given in Table 1 [22].

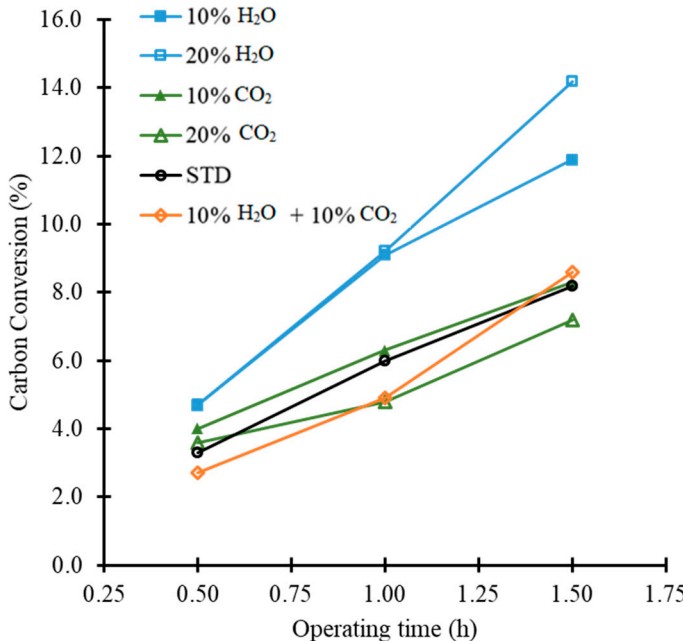

**Figure 7.** Impact of gas composition on carbon conversion during the operating time. T = 850 °C; $\tau$ = 1.2 s; $d_p$ = 500–630 μm.

*2.5. Impact of Char Particle Size*

To evaluate the char catalytic activity, it is important to test it under conditions with no heat and mass transfer limitations. Several criteria can be used to test experimentally these limitations as explained by Madon et al. [23]. However, our previous work [14] considered such parameters and modeled the regions of their effect. In this section, the char particle size was the only parameter tested. Figure 8 shows the impact of the char particle size on naphthalene conversion at 750 °C. It was found that doubling the average particle size reduced the naphthalene conversion by only 5.5 wt.%. Therefore, it is difficult to confirm that the internal mass transfer limitations were negligible. However, we can more accurately say that there were no internal mass transfer limitations in the larger pores.

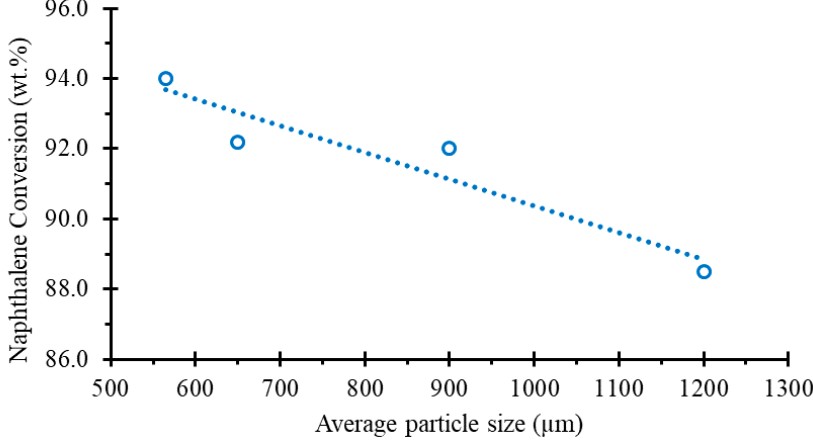

**Figure 8.** Impact of particle size on naphthalene conversion. T = 750 °C; t = 15 min.

The impact of particle size on carbon conversion was tested as shown in Figure 9. In this test, two char particle sizes were used: 500–630 μm and 1400–1700 μm. As found with naphthalene conversion, only a limited effect on carbon conversion (<1%) was attained. Therefore, it is also difficult to confirm here that the internal mass transfer limitations were negligible. However, it may be more accurate to say that there were no internal mass transfer limitations in the larger pores.

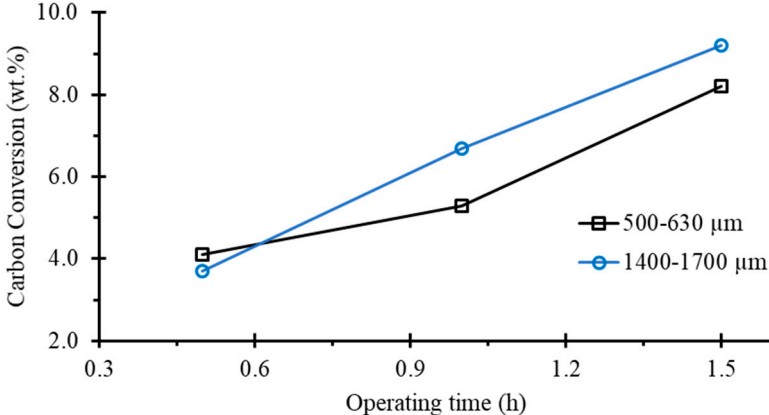

**Figure 9.** Impact of particle size on carbon conversion during the operating time. T = 850 °C; τ = 1.2 s.

### 2.6. Impact of Char Surface Area

Both the surface structure and chemical composition of the catalyst influences the catalyst activity. Thus, the impact of the char surface area on naphthalene conversion was studied. The char surface area was measured as explained in Section 3.1. Furthermore, the char surface properties were manipulated by producing the pinewood-char at different pyrolysis heating rates (10, 25, and >50 °C /min) and final pyrolysis temperatures (700, 900, and 950 °C) as shown in Figure 10. It was found that the highest naphthalene conversion (99.9 wt.%) was attained at 25 °C/min heating rate and 700 °C final pyrolysis temperature. However, it was difficult to draw conclusions on the impact of surface area at such high naphthalene conversions. On the other hand, these selected conditions were often used in hot gas catalytic cleaning processes.

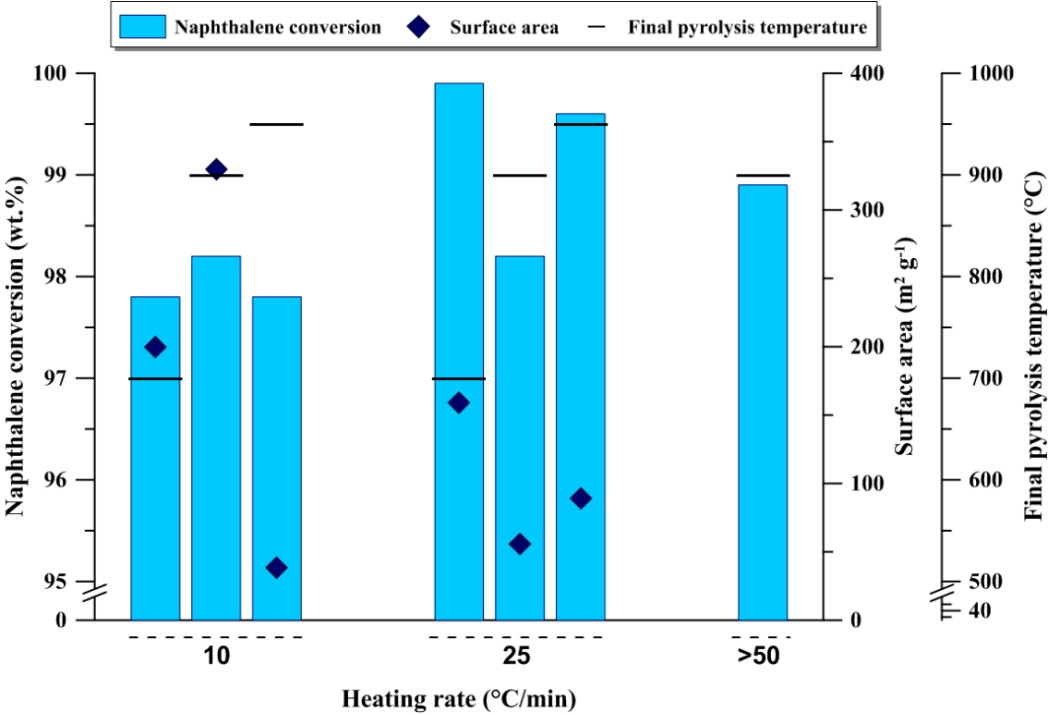

**Figure 10.** Impact of pyrolysis conditions on char surface area and naphthalene conversion. τ = 0.25 s; $d_p$ = 100–500 μm.

### 2.7. Impact of Char Source

The char catalytic activity is affected, besides the surface area, by the chemical composition. The mineral content of the char promotes the tar and char gasification reactions. In order to investigate this influence, char was prepared from three types of precursors: pinewood biomass, rietspruyt coal, and brown coal. Table 4 gives the chemical composition of the tested char. It can be observed that the coal-char have higher ash and iron oxide content than the biomass char.

**Table 4.** The chemical composition of tested char (mass fractions).

|  | **Biomass** | **Coal** | |
|---|---|---|---|
|  | **Pinewood** | **Rietspruyt** | **Brown** |
| C | 87.9% | 82.3% | 89.6% |
| N | 0.3% | 2.0% | 0.8% |
| H | 0.6% | 0.4% | 1.1% |
| Ash | 4.7% | 15.3% | 7.8% |
| O (by difference) | 6.5% | 0.0% | 0.7% |
| MgO | 0.203% | 0.285% | 0.086% |
| $Al_2O_3$ | - | 4.700% | 0.318% |
| $K_2O$ | 0.846% | 0.059% | 0.006% |
| CaO | 1.810% | 0.793% | 2.353% |
| $Fe_2O_3$ | 0.114% | 0.275% | 0.852% |

Figure 11 shows that the char from coal sources gave higher naphthalene conversion than the pinewood char. Furthermore, the brown char has the highest naphthalene conversion. This can be associated with the higher ash content than pinewood-char. It can also be related to the highest iron-oxide content, which speeds up carbon gasification reactions [10]. Therefore, special attention should be given to the iron content.

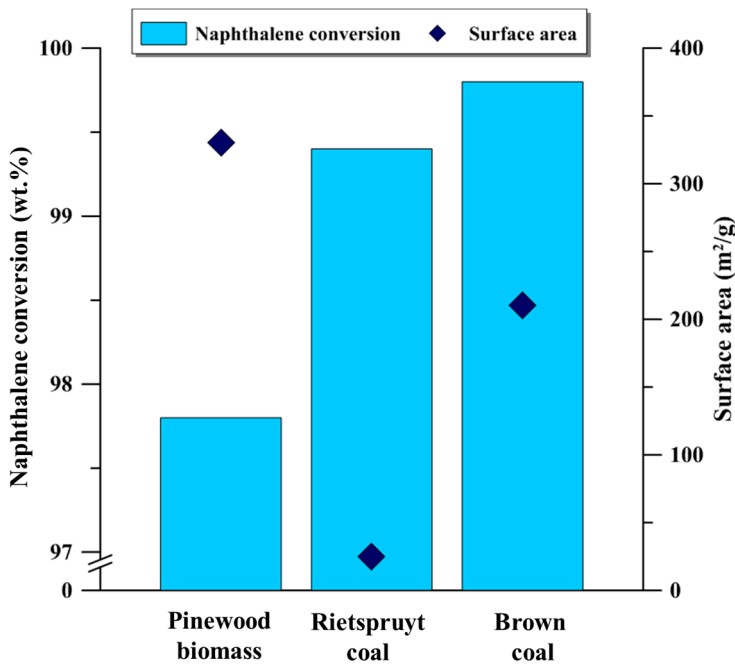

**Figure 11.** Impact of char source on surface area and naphthalene conversion. $\tau = 0.25$ s; $d_p = 100$–$500$ μm.

## 3. Materials and Methods

A set of experiments were carried out in a macro-TGA fixed bed reactor setup. The goal was to investigate the influence of some key parameters on tar conversion using biomass char. Naphthalene was selected as a tar model component as explained in a previous study [16].

### 3.1. Biomass Char Characterization

The biomass char was sourced from a commercial supplier. It was pre-treated by heating up to 850 °C and then soaking for 30 min (duration of char final pyrolysis temperature). The proximate, ultimate, and ash analysis of treated biomass char were analyzed as shown in Table 5. The surface area and porosity of the biomass char were measured by nitrogen adsorption at 77 K. Moreover, the specific surface area was estimated using the Brunauer–Emmett–Teller (BET) method [24]. The biomass char contains mainly carbon (89 wt.%) with effective ash content (9.55 wt.%). The ash metals, especially the alkaline earth metals (Ca, Mg) and iron (Fe) are the highest and can play an important role in promoting the naphthalene reforming reactions.

**Table 5.** Proximate, ultimate, and ash analysis of treated biomass char.

| Analysis | Composition | Mass Fraction |
|---|---|---|
| Proximate analysis | Fixed carbon | 88.24% |
| | Ash | 9.55% |
| | Volatiles | 2.01% |
| | Water | 0.20% |
| Ultimate analysis | C | 89.03% |
| | O* | 10.00% |
| | F | 0.40% |
| | N | 0.24% |
| | H | 0.12% |
| | Cl | 0.02% |
| | S | 0.01% |
| | Br | 0.01% |
| Ash analysis | Ca | 29.90% |
| | Mg | 12.40% |
| | Fe | 9.10% |
| | Al | 2.21% |
| | Na | 1.09% |
| | Ti | 0.81% |
| | Si | 0.66% |
| | K | 0.48% |
| | C | 0.17% |

\* by difference

Table 6 shows the surface area characteristics of the tested biomass char. The surface area of the biomass char in this work was comparable with the results of Park et al. [8] who achieved 320 $m^2 \cdot g^{-1}$ BET for char produced from wood pyrolysis at 800 °C. However, it is worth mentioning that the pyrolysis temperature and precursor material affect the surface characteristics of the produced char.

**Table 6.** Surface characteristics of tested biomass char.

| Property | Value | Unit |
|---|---|---|
| $N_2$–BET | 353 | $m^2 \cdot g^{-1}$ |
| Pore volume | 0.19 | $cm^3 \cdot g^{-1}$ |
| Average pore width | 2.9 | nm |

### 3.2. Setup

A simple macro-TGA fixed bed reactor setup was used in these experiments as shown in Figure 12. A gas feeding system used to produce a feed gas to the reactor consisted of $CO_2$, $CO$, $H_2$, $N_2$, $H_2O$, and naphthalene. Three lines were mixed to form this composition: (1) gas line consisting mainly of $CO_2$, $CO$, $H_2$, $N_2$, (2) saturated nitrogen with steam, and (3) saturated nitrogen with naphthalene. The saturated nitrogen streams were passed through two saturation units that contained heated water and melted naphthalene. The concentration of both components at the exit nitrogen stream was controlled by the temperatures of the liquids in the saturation units and nitrogen flow rates. The gas lines containing water and naphthalene were heated to 250 °C and insulated to prevent any condensation.

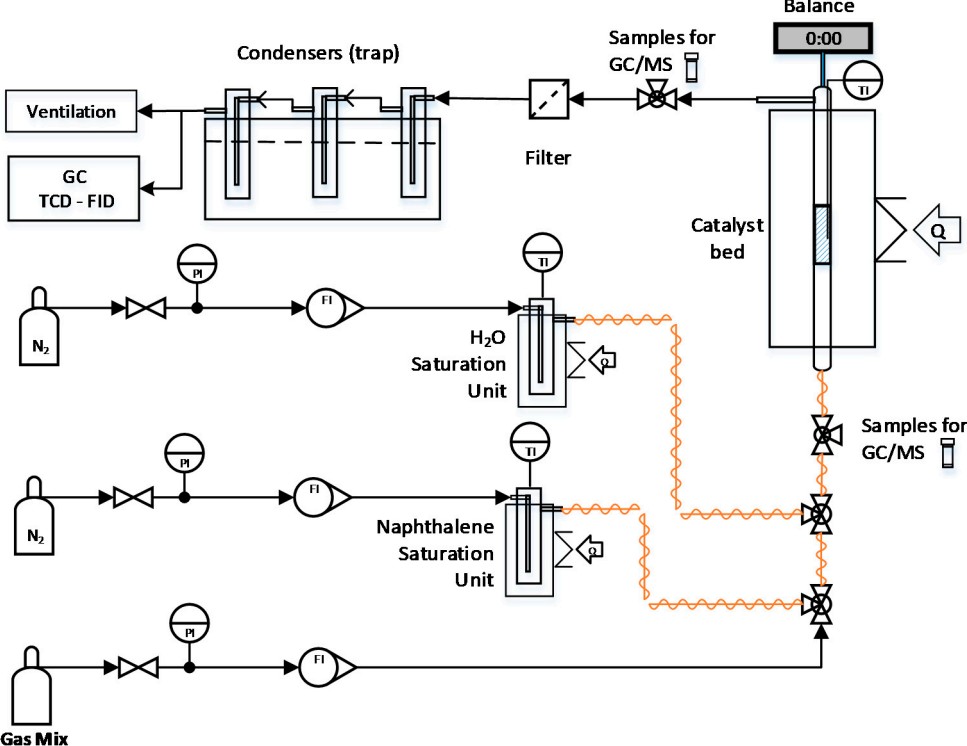

**Figure 12.** Macro thermogravimetric fixed bed experimental setup.

The fixed bed tubular reactor was made of quartz. The height of the tube is 75 cm and the internal diameter is 4 cm. The tube is located in a tubular electrical furnace. A porous quartz disc was installed in the middle of the tubular reactor to hold the catalyst bed. The top of the reactor was suspended on a balance to measure its weight at different operating times (time on stream), which mimics the instrument of thermal gravimetric analysis (TGA). However, the balance in this setup had the following differences: (1) the monitored weight was not only the char bed but the whole reactor weight, (2) a less sensitive balance was used instead of a microbalance, (3) the weight was not measured continuously but at certain operating times, and (4) the reactor weight was measured by stopping the gas feed flow, disconnecting the top and bottom of the reactor, and hanging the top of the reactor manually on the balance hook placed above the reactor.

The initial weight of the char bed was measured at room temperature. The reactor was then heated to the required temperature under nitrogen flow. It was important to monitor the change in reactor weight due to loss of moisture and pyrolysis vapors originating from the char bed. The resulting weight of the char bed at the desired reactor temperature was used as the starting weight of char bed for naphthalene conversion.

The naphthalene conversion was measured by taking samples from the sampling points at the reactor's in- and outgoing gas using the solid phase adsorption (SPA) method [25]. The analysis of samples was done offline with a gas chromatograph-mass spectrometer (GC-MS).

The conversion of carbon was measured by weighing the reactor at the reactor temperature and variable operating time. The conversion was calculated as follows:

$$X_{c,t} = \frac{w_o - w_t}{w_c \cdot w_O} \tag{1}$$

where $X_{c,t}$ is the time-based carbon conversion, $w_o$ is the initial char weight, $w_t$ is the time-based char weight, and $w_c$ is the carbon content of the treated char.

### 3.3. Test Conditions

The experiments were conducted to study the catalytic tar conversion activity and the concurrent carbon conversion of the biomass char. The test conditions are given in Table 2.

### 4. Conclusions

In this paper, the impact of several parameters on naphthalene and char conversion was tested in a macro-TGA fixed char bed reactor. Under the tested conditions, the following conclusions are drawn:

(1) Biomass char was found to be a highly active catalyst. However, its catalytic activity depended on the bed temperature and operating time. Therefore, any application for biomass char as a catalyst should consider the compensation of the weight loss from biomass char generation in the gasifier. A temperature in the range of 800–850 °C seems to be optimal for high tar conversion and limited concurrent carbon conversion.

(2) Naphthalene conversion reactions involved cracking reactions that produced coke, which reduced the catalytic activity of char for naphthalene conversion and reduced the reactivity of char in the gasification reactions.

(3) Minor internal mass transfer limitations in the larger pores of the char affected the naphthalene and char conversion reactions.

(4) High ash and iron contents in the char increased the char activity.

**Author Contributions:** Conceptualization, Z.A.E.-R., E.B., and G.B.; Methodology, Z.A.E.-R., E.B., and G.B.; Validation, Z.A.E.-R.; Formal Analysis, Z.A.E.-R.; Investigation, Z.A.E.-R.; Resources, Z.A.E.-R., E.B., and G.B.; Data Curation, Z.A.E.-R., and S.A.-G.; Writing-Original Draft Preparation, Z.A.E.-R.; Writing-Review and Editing, Z.A.E.-R., and S.A.-G.; Visualization, Z.A.E.-R., and S.A.-G.; Supervision, E.B., and G.B.; Project Administration, Z.A.E.-R.; Funding Acquisition, G.B.

**Funding:** This work was carried out within the framework of the Center of Separation Technology of the University of Twente. This center is financed by the Netherlands Organization for Applied Scientific Research (TNO) and the Institute of Mechanics, Processes, and Control-Twente (IMPACT).

**Conflicts of Interest:** The authors declare no conflict of interest.

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
