# Peer review of "Impact of Char Properties and Reaction Parameters on Naphthalene Conversion in a Macro-TGA Fixed Char Bed Reactor"

_catalysts, doi:10.3390/catal9040307_

Round 1

Reviewer 1 Report

Comments to Ziad Abu el-Rub et al. – catalysts-457497

Summary

The authors present an experimental study related to tar removal in biomass gasification, in particular the catalytic performance of biomass char and with naphthalene as an example of a removable tar component. In order to establish optimal conditions for the naphthalene conversion, taking into account the preservation of the catalytic activity of the char, the researchers conducted experiments at different temperatures and with different char particle sizes, naphthalene concentrations, feed gas compositions etc.

General comments

The topic of the study is well within the realm of the Catalysts journal and the results, although not being very surprising, could still be of practical interest to the Catalysts journal readers. Furthermore, the manuscript contains the customary sections of a scientific journal article, i.e. Introduction, Materials and Methods, Results and Discussion as well as Conclusions. Of these, the Introduction section contains adequate information on previous studies related to the topic, but the authors could present their own objectives in a more structured way. As for the Materials and Methods section, it is clear and detailed enough for other researchers to repeat the experiments. The Results and Discussion section, in its turn presents the experimental results comprehensively, but the discussion part would benefit from some more careful explanation. Finally, the Conclusions section could present the main findings of the study in a more organized way, e.g. as a list.

Obviously, English is not a problem for the authors, but some of the sentences would be clearer with another order of the subordinate clauses.

Detailed comments

Lines 12-13:     I think the first sentence would easier to understand this way: Catalytic tar removal is one of the main challenges for the successful commercialization of biomass gasification.

Line 16:                                           … such a catalyst …

Line 24:             Furthermore

Line 36:             Is this a reasonable definition of biomass? Fossil fuels, such as coal and mineral oil also have plant and animal origin dating e.g. from the carboniferous period.

Lines 39-40:     I have some problems with this word order as well. Do you mean: Among these technologies gasification, which converts biomass into combustible producer gas, is considered one of the preferable options.

Lines 49-50:     The last method, where a catalyst is applied under a temperature that essentially matches that of the gasifier, is considered the most economical one.

Line 63:             Is here a word missing? … and their subsequent …

Line 64:                                           … remove a hydrogen atom …

Line 70: Using the word further in this way is rather frequent but not quite correct; when you are introducing a new argument, it should be furthermore (= additionally). You can use further (= additional) to modify a noun e.g. further char deactivation was observed.

Lines 89-90:     Not a complete sentence.

Line 104:           Reference for the BET-method.

Line 159:           … test conditions are given …

Lines 173-174:                              The whereas sentence as well as the following one are a bit irregular. Should it be even lower naphthalene conversion?

Lines 188-190:                              The meaning of the sentence is obscure. Should it be: In contrast, the increase of the naphthalene conversion might be related to the net effect etc.? The latter part of the sentence would however still be murky. Better to explain carefully what you mean in a few sentences.

Lines 210-211:                              According to Figure 6, the temperature is also higher in the experiment with longer residence time. Is it then possible to tell whether it is the temperature or the residence time that is significant?

Lines 235-236:                              The reaction could still be e.g. second order, if the reaction speed were proportional to the square of the remaining char.

Line 307:           Furthermore or moreover.

Line 337:           Colon instead of semicolon. I think you could also omit the punctuation.

Line 361:           Furthermore

Lines 423-424:                             … the conducted experiments …

Line 425:           … alkali content in the char was relevant …

Author Response

Response to Reviewer 1:

The authors would like to thank the reviewer for the precious time and important comments. We have carefully addressed all the comments. The corresponding changes and modifications made in the revised paper are summarized below. Our responses are given in a point-by-point manner below each comment in blue color. Changes to the manuscript are shown below the response to each comment in green color.

General comments

 The topic of the study is well within the realm of the Catalysts journal and the results, although not being very surprising, could still be of practical interest to the Catalysts journal readers. Furthermore, the manuscript contains the customary sections of a scientific journal article, i.e. Introduction, Materials and Methods, Results and Discussion as well as Conclusions. Of these, the Introduction section contains adequate information on previous studies related to the topic, but the authors could present their own objectives in a more structured way. As for the Materials and Methods section, it is clear and detailed enough for other researchers to repeat the experiments. The Results and Discussion section, in its turn presents the experimental results comprehensively, but the discussion part would benefit from some more careful explanation. Finally, the Conclusions section could present the main findings of the study in a more organized way, e.g. as a list.

Obviously, English is not a problem for the authors, but some of the sentences would be clearer with another order of the subordinate clauses

Response:

·         The objective was rephrased as per the reviewer suggestion.

·         The authors considered the all reviewers’ and editors comments, which improved the discussion part

·         The authors revised the conclusions section where the main findings were listed as per the reviewer suggestion

Changes to manuscript

Line 101-106

The objective of this work is to study the impact of secondary measures on naphthalene and char conversion in a macro-TGA fixed char bed reactor. These measures include the impact of reaction parameters and char properties. The studied parameters were char bed temperature, gas residence time in the char bed, char particle size, feed naphthalene concentration, feed gas composition, char surface area, and char precursor. The outcome of this research will help in the design of an integrated process that utilizes the byproduct biomass char as a catalyst in the gasification process.

Line 337-349

In this paper, the impact of several parameters on naphthalene and char conversion was tested in a macro-TGA fixed char bed. Under the tested conditions, the following conclusions are drawn:

1)       Biomass char was found to be a highly active catalyst. However, its catalytic activity depended on the bed temperature and operating time. Therefore, any application for biomass char as a catalyst should consider the compensation of the weight loss from biomass char generation in the gasifier. A temperature in the range of 800-850 °C seems to be optimal for high tar conversion and limited concurrent carbon conversion.

2)       Naphthenic conversion reactions involve cracking reactions that produce coke, which reduces the catalytic activity of char for naphthalene conversion, and the reactivity of char in gasification reactions.

3)       Minor internal mass transfer limitations in the larger pores of the char affected the naphthalene and conversion reactions.

4)       High ash and iron contents in the char increased the char activity.

 Detailed comments

Lines 12-13:     I think the first sentence would easier to understand this way: Catalytic tar removal is one of the main challenges for the successful commercialization of biomass gasification.

Response: Agree

Changes to manuscript: [Line 13-14]

Catalytic tar removal is one of the main challenges for the successful commercialization of biomass gasification.

Line 16:                                           … such a catalyst …

Response: Agree. The article “a” was added

Changes to manuscript: [line 16  ]

Biomass char has the potential to be such a catalyst

Line 24:             Furthermore

Response: Agree. Further was changed to Furthermore

Changes to manuscript: [line 24 ]

Furthermore, high ash and iron contents enhanced char activity

Line 36:             Is this a reasonable definition of biomass? Fossil fuels, such as coal and mineral oil also have plant and animal origin dating e.g. from the carboniferous period.

Response: Agree. The definition was made more specific

Changes to manuscript: [line 36-37]

 Biomass is a generic term for organic materials often come from living or recently died plants and animals.

Lines 39-40:     I have some problems with this word order as well. Do you mean: Among these technologies gasification, which converts biomass into combustible producer gas, is considered one of the preferable options.

Response: Agree. The sentence was rephrased

Changes to manuscript: [line 39-41]

Several chemical-thermal technologies can be used to exploit the energy stored in biomass. Gasification is one of the notable technologies, where biomass is converted by partial combustion into a combustible gas.

Lines 49-50:     The last method, where a catalyst is applied under a temperature that essentially matches that of the gasifier, is considered the most economical one.

Response: Agree. The suggested change was made.

Changes to manuscript: [line 48-49]

The last method, where a catalyst is applied under a temperature close to that of the gasifier, is considered the most economical one [5]

Line 63:             Is here a word missing? … and their subsequent …

Response: Yes. The word “and” was added

Changes to manuscript: [line 63]

Fuentes-Cano et al. [9] found that the char catalytic activity is strongly dependent on the concentration of alkali metals (e.g. K and Na) and alkaline earth metals (e.g. Ca). Furthermore, they reported that the absence of these metals caused a quick char deactivation

Line 64:                                           … remove a hydrogen atom …

Response: the word atom was removed , and the discussion becomes on the hydrogen in general.

Changes to manuscript: [line 68]

Greensfelder et al. [11] reported that the biomass char converts the tar into free radical by removing the hydrogen.

Line 70: Using the word further in this way is rather frequent but not quite correct; when you are introducing a new argument, it should be furthermore (= additionally). You can use further (= additional) to modify a noun e.g. further char deactivation was observed.

Response: Agree. A modification to the paragraph was made and the word “further” was not needed.

Changes to manuscript: [line 74]

Nestler et al. [12] tested the decomposition of naphthalene over different wood chars. They found that naphthalene was completely converted to carbon and hydrogen, as they did not detect by the GC other gaseous components except hydrogen.

 Lines 89-90:     Not a complete sentence.

Response: Agree. The paragraph  was corrected and removed.

Changes to manuscript: [line 101-106]

The objective of this work is to study the impact of secondary measures on naphthalene and char conversion in a macro-TGA fixed char bed reactor. These measures include the impact of reaction parameters and char properties. The studied parameters were char bed temperature, gas residence time in the char bed, char particle size, feed naphthalene concentration, feed gas composition, char surface area, and char precursor. The outcome of this research will help in the design of an integrated process that utilizes the byproduct biomass char as a catalyst in the gasification process.

Line 104:           Reference for the BET-method.

Response: Agree. The following reference was added: Brunauer, S.; Emmett, P.H.; Teller, E. Adsorption of gases in multimolecular layers. J. Am. Chem. Soc. 1938, 60, 309-319 DOI: 10.1021/ja01269a023.

Changes to manuscript: [line 117]

Moreover, the specific surface area was estimated using the BET method [20].

Line 159:           … test conditions are given …

Response: Agree. Were is changed to are

Changes to manuscript: [line 170]

The test conditions are given in Table 4

Lines 173-174:                              The whereas sentence as well as the following one are a bit irregular. Should it be even lower naphthalene conversion?

Response:

Agree. This is a typo error, and the sentence is corrected.

Changes to manuscript: [line   ]

Whereas, Park et al. [8] found 75% maximum primary tar conversion in the gasifier at 800 °C using hot char particles. In addition, Nestler et al. [12] found also lower naphthalene conversion (76%) at 850 oC for wood char pyrolyzed at 500 and 800 oC.

Lines 188-190:                              The meaning of the sentence is obscure. Should it be: In contrast, the increase of the naphthalene conversion might be related to the net effect etc.? The latter part of the sentence would however still be murky. Better to explain carefully what you mean in a few sentences.

Response:

Agree. The whole paragraph is corrected.

Changes to manuscript: [line 191-199]

On the other hand, the naphthalene conversion at 750 °C decreased with time from 96% to 74% at the 6th hour then increased to 84% at the 8th hour.  This can be related to the char lower activity at this temperature resulted from the coke formation, which reduces the access to the active surface area of the char. The catalytic activity of the char particles are affected on one hand by the char activation resulted in carbon loss because of its reaction with steam and carbon dioxide. This activation exposes more active sites and surface metals that promote reforming reactions of naphthalene. On the other hand, the char is deactivated by the coke formation resulted from the naphthalene cracking reactions. Therefore, the later increase of the naphthalene conversion after the 6th hour might be related to the positive net effect. 

Lines 210-211:                              According to Figure 6, the temperature is also higher in the experiment with longer residence time. Is it then possible to tell whether it is the temperature or the residence time that is significant?

Response: Agree. The whole paragraph is clarified

Changes to manuscript: [line 216-218]

At 850 °C and longer gas residence time (2.4 s), the carbon conversion increased to 27 wt.% after 5.7 hours operating time. However, it was better to vary one parameter only (temperature) and fix the value of gas residence time because both of them have a positive influence on carbon conversion.

Lines 235-236:                              The reaction could still be e.g. second order, if the reaction speed were proportional to the square of the remaining char.

Response:

Agree with reviewer comment. However, the paragraph was further explained to add more information that explain our result.

Changes to manuscript: [line 236-245]

Naphthalene conversion reactions include cracking reactions that produce coke, which deactivates the activity of the char. Therefore, the impact of feed naphthalene loading on its conversion was tested. The test was conducted at 6 to 18 g∙Nm^(-3) feed naphthalene concentration, 0.15 s gas residence time, and 900 °C, 1400-1700 µm char particle size, 15 min operating time. There was no significant effect of the feed naphthalene concentration as the conversion was almost constant (89 wt.%-91 wt.%). This showed that the naphthalene-char reaction was first order because: (i) it was independent of the initial naphthalene concentration at these process conditions, (ii) no significant char consumption occurred during the studied operating time, (iii) most heterogeneous gas phase reactions are first order. However, for longer operating times, the deposition of coke resulted from naphthalene cracking reactions can cause a serious decline in the char activity.

Line 307:           Furthermore or moreover.

Response: Agree. There was rephrasing for the paragraphs and this word “further” was removed

Line 337:           Colon instead of semicolon. I think you could also omit the punctuation.

Response: Agree. There was rephrasing for the paragraphs and semicolon was removed.

Line 361:           Furthermore

Response: Agree. The word further is changed to furthermore

Changes to manuscript: [line309]

Lines 423-424:                             … the conducted experiments …

Response: Agree. There was rephrasing for the paragraphs and word was not needed anymore.

Line 425:           … alkali content in the char was relevant …

Response:

Response: Agree. There was rephrasing for the paragraphs and word was not needed anymore

Reviewer 2 Report

The current manuscript reports on the impact of char properties and reaction parameters on naphthalene conversion in a macro-TGA fixed char. The work contained in this manuscript is weak and it offers very little new insights into the pathways for a removal of model tar component, naphthalene. The recommendation is that the current version of this manuscript is not suitable for publication in Catalysts

Author Response

Response to Reviewer 2

The authors would like to thank the reviewer for the precious time and important comments. We have carefully addressed all the comments. The corresponding changes and modifications made in the revised paper are summarized below. Our responses are given in a point-by-point manner below each comment in blue color. Changes to the manuscript are shown below the response to each comment in green color.

Comments and Suggestions for Authors

The current manuscript reports on the impact of char properties and reaction parameters on naphthalene conversion in a macro-TGA fixed char. The work contained in this manuscript is weak and it offers very little new insights into the pathways for removal of model tar component, naphthalene. The recommendation is that the current version of this manuscript is not suitable for publication in Catalysts

Response:

The authors value the scientific opinion of the reviewer.  The revised version of the manuscript focused on giving more insight into the reactions, kinetics, and pathways for naphthalene and char conversion, especially in the introduction section. It also directed the reader to previous work for the authors on a   developed model that studied the activity of the biomass char particle, naphthalene conversion, and the concurrent carbon conversion of the particle.  Moreover, the many paragraphs in the discussion of the manuscript were revised.

Changes to manuscript

[line 75-106]

Figure 1 shows a scheme of the homogenous and heterogeneous reactions that occur on the surface of the char particle and the atmosphere of its producer gas. There is mainly one homogenous gas phase reaction, which is the water-gas shift reaction. Whereas, the biomass char particle undergoes several heterogeneous gasification reactions, which mainly are the water gas reaction ( and the Boudouard reaction (. On one hand, the tar is subjected to other heterogeneous reactions, which are mainly the steam reforming (tar + steam) and dry reforming (tar + CO2) reactions. These reactions are activated by the active sites on the char surface to produce CO and H2. However, they might produce also reactive coke that is consumed by these reforming reactions [13]. On the other hand, tar can be adsorbed on the char surface to form free radicals by removing the hydrogen. These free radicals can enter heavy polymerization reaction to produce less reactive coke, which deactivates the char surface active sites [13].

Figure 1. Scheme of the homogenous reaction (water-gas shift reaction) and qualitative (unbalanced) heterogeneous reactions (tar polymerization, tar reforming, char gasification, coke gasification) on the surface and atmosphere of the char particle.

In a previous work [14], a single particle model of biomass char was developed. This model studied the activity of the biomass char particle, naphthalene conversion, and the concurrent carbon conversion of the particle. Table 1 summarizes the main reactions considered. It was found that there was no significant effect for internal or external mass transfer limitations on the naphthalene and carbon conversion. However, the conversion reactions affected the physical properties and pore structure of the char particle. Furthermore, the char particle could be assumed isothermal up to a bulk temperature of 1160 °C.

Table 1. Main reactions occur in a fixed char bed reactor for treating naphthalene as a model tar component. 

Reviewer 3 Report

I can recommend this article for publication after major revision.

Title, ‘’a macro-TGA fixed char bed’’, I am not sure about this terminology, it must be as ’’a macro-TGA fixed bed reactor’’. It this is the accurate terminology, authors have to correct it in several places in the manuscript.

Take care of upper cases in middle of the sentence e.g., Naphthalene, line 21

Line 37, ‘’one of these promising renewable energy sources’’ do you mean ‘’wind, geothermal, solar, hydropower, and biomass’’ (Chem. Soc. Rev., 2018, 47, 8349-8402)

Line 62, ‘’Nzihou et al. [8] related their catalytic influence’’ do you mean alkali and alkaline earth metal based catalysts. Specify it.

As well, modify the following sentence, in which ‘’their’’ is repeated at least three times and it is not clear what they want to reveal here ‘’ Nzihou et al. [8] related their catalytic influence to their 63 release at elevated temperatures their subsequent dispersion in the carbon matrix’’

Line 70, ‘’converted to carbon’’ do you mean CO and CO2?, make it clear.

Line 72, ‘’reaction type’’, do you mean reactor type?

Fig. 1, take care of subscripts and it is not clear which reaction belongs to homogeneous or heterogeneous. Make it clear. Also, balance the equations.

Introduction - a major concern is the lack of latest literature refs. The following review article can provide recent advances in biomass valorization as well as the application of biomass-derived catalysts including biomass char and these review articles can be referred in the introduction part.

Chem. Soc. Rev., 2018, 47, 8349-8402 and DOI:10.1039/C8CS00452H

Line 102, ‘’soaking for 30 minutes’’, which solvent was used for soaking?

Fig. 2, take care of subscripts in N2 and H2O

Table 3, 1.01∙105

Line 170, insert ‘’respectively’’ after 900 °C

Line 172, modify the following sentence, in which ‘’found’’ is repeated

Park et al. [6] found lower naphthalene conversion found that

take care of subscripts in Figure 8.

Line 352, 330 ?2∙?−1 (see Table 1), but Table 1 shows proximate, ultimate, ash analysis of treated biomass char and also it it 353 ?2∙?−1 as mentioned in Table 2.

Typo corrections

tars to tar, line 48

hydrogen atom to hydrogen, line 64

Bartholomew to Bartholomew et al., line 72

cokes to coke, line 72

Tars are to tar is, line 78

Chars to char, line 103

Reduction to conversion, line 182, 185

Take care of chars, tars, and cokes to char, tar and coke throughout the manuscript.

Author Response

Response to Reviewer 3

The authors would like to thank the reviewer for the precious time and important comments. We have carefully addressed all the comments. The corresponding changes and modifications made in the revised paper are summarized below. Our responses are given in a point-by-point manner below each comment in blue color. Changes to the manuscript are shown below the response to each comment in green color.

I can recommend this article for publication after major revision.

Title, ‘’a macro-TGA fixed char bed’’, I am not sure about this terminology, it must be as ’’a macro-TGA fixed bed reactor’’. It this is the accurate terminology, authors have to correct it in several places in the manuscript.

Response: Agree. Adding the word “reactor” to “fixed bed” gives better description

Changes to manuscript: [Line 4]

Impact of char properties and reaction parameters on naphthalene conversion in a macro-TGA fixed char bed reactor

Take care of upper cases in middle of the sentence e.g., Naphthalene, line 21

Response: Agree. The first letter of naphthalene is changed to small letter (found only correction, which is the one pointed out by the reviewer).

Changes to manuscript: [line 21]

(CO, CO2, H2O, H2, CH4, naphthalene, and N2)

Line 37, ‘’one of these promising renewable energy sources’’ do you mean ‘’wind, geothermal, solar, hydropower, and biomass’’ (Chem. Soc. Rev., 2018, 47, 8349-8402)

Response:

Yes.   The examples and the and the given reference were added.

Changes to manuscript: [line 38-39  ]

It is considered one of these promising renewable energy sources, such as solar, wind, geothermal, and hydropower [2].

Line 62, ‘’Nzihou et al. [8] related their catalytic influence’’ do you mean alkali and alkaline earth metal based catalysts. Specify it.

As well, modify the following sentence, in which ‘’their’’ is repeated at least three times and it is not clear what they want to reveal here ‘’ Nzihou et al. [8] related their catalytic influence to their 63 release at elevated temperatures their subsequent dispersion in the carbon matrix’’

Response: Agree. The paragraph was made clearer.

Changes to manuscript: [line  62-66]

Fuentes-Cano et al. [9] found that the char catalytic activity is strongly dependent on the concentration of alkali metals (e.g. K and Na) and alkaline earth metals (e.g. Ca). Furthermore, they reported that the absence of these metals caused a quick char deactivation. However, Nzihou et al. [10] related the catalytic influence of the alkali and alkaline metals to their release at elevated temperatures and the subsequent dispersion of these metals in the carbon matrix.

Line 70, ‘’converted to carbon’’ do you mean CO and CO2?, make it clear.

Response:

No. it means as said, naphthalene was completely converted to carbon and hydrogen. Nestler found that the analysis of the reactor off-gas composition showed that the converted naphthalene was completely decomposed into carbon and hydrogen since no other gaseous components except for hydrogen were detected by the GC.

Changes to manuscript: [line 72-74 ]

 Nestler et al. [12] tested the decomposition of naphthalene over different wood chars. They found that naphthalene was completely converted to carbon and hydrogen, as they did not detect by the GC other gaseous components except hydrogen.

Line 72, ‘’reaction type’’, do you mean reactor type?

Response:

No. it means reaction type as reported by Bartholomew [14]. This because coke is produced by decomposition or condensation of hydrocarbons on catalyst surfaces and typically consists of polymerized heavy hydrocarbons. The types of produced hydrocarbons and gases differ according to the reaction type. However, the whole paragraph was revised.

Changes to manuscript [line75-85]

 Figure 1 shows a scheme of the homogenous and heterogeneous reactions that occur on the surface of the char particle and the atmosphere of its producer gas. There is mainly one homogenous gas phase reaction, which is the water-gas shift reaction. Whereas, the biomass char particle undergoes several heterogeneous gasification reactions, which mainly are the water gas reaction ( and the Boudouard reaction (. On one hand, the tar is subjected to other heterogeneous reactions, which are mainly the steam reforming (tar + steam) and dry reforming (tar + CO2) reactions. These reactions are activated by the active sites on the char surface to produce CO and H2. However, they might produce also reactive coke that is consumed by these reforming reactions [13]. On the other hand, tar can be adsorbed on the char surface to form free radicals by removing the hydrogen. These free radicals can enter heavy polymerization reaction to produce less reactive coke, which deactivates the char surface active sites [13].

Fig. 1, take care of subscripts and it is not clear which reaction belongs to homogeneous or heterogeneous. Make it clear. Also, balance the equations.

Response:

Agree. Figure 1 is modified according to the reviewer comments and made clearer: (1) Subscripts were made, (2) more colors were added to clarify the scheme, (3) in the figure name, heterogeneous and homogenous reactions were specified.  Heterogeneous reactions are mentioned in the figure title as unbalanced as their balance depends on the molecular formula of the coke, tar, and char. Further, they can be a summation of more than one reaction.

Changes to manuscript: [line  84]

Figure 1. Scheme of the homogenous reaction (water-gas shift reaction) and qualitative (unbalanced) heterogeneous reactions (tar polymerization, tar reforming, char gasification, coke gasification) on the surface and atmosphere of the char particle.

Introduction - a major concern is the lack of latest literature refs. The following review article can provide recent advances in biomass valorization as well as the application of biomass-derived catalysts including biomass char and these review articles can be referred in the introduction part.

Chem. Soc. Rev., 2018, 47, 8349-8402 and DOI:10.1039/C8CS00452H

Response: Agree. Referring to this review paper was added to the introduction

Changes to manuscript: [line 57-58]

Sudarsanam et. al. [7] presented a comprehensive review on recent advances in catalytic biomass conversions in addition to applications of a biomass-derived catalyst including biomass char.  

Line 102, ‘’soaking for 30 minutes’’, which solvent was used for soaking?

Response: No solvent was used. The soaking time represents the residence time of char at the final pyrolysis temperature.

Changes to manuscript: [line 113-114]

The biomass char was sourced out from a commercial supplier. It was pre-treated by heating up to 850 and then soaking for 30 minutes (duration of char final pyrolysis temperature).

Fig. 2, take care of subscripts in N2 and H2O

Response:

Agree. Numbers were converted to subscripts in Figure 2 and a box for gas analysis was added.

Changes to manuscript: [line 160 ]

 Figure 2. Macro thermogravimetric fixed bed experimental setup

Table 3, 1.01∙105

Response: Not clear what the reviewer meant. In the manuscript, it is written correctly

Changes to manuscript: None.

Line 170, insert ‘’respectively’’ after 900 °C

Response: Agree.

Changes to manuscript: [line 178]

The naphthalene conversion at 15 min time on stream was in the range of 70% and 100 % in the temperature range of 700 and 900 °C, respectively.

Line 172, modify the following sentence, in which ‘’found’’ is repeated

Park et al. [6] found lower naphthalene conversion found that

Response:

Agree.

Changes to manuscript: [line 180-181]

Whereas, Park et al. [8] found 75% maximum primary tar conversion in the gasifier at 800 °C using hot char particles.

take care of subscripts in Figure 8.

Response: Agree. Subscripts were corrected.

Changes to manuscript: [line 283-285]

Figure 8. Impact of gas composition on carbon conversion during the operating time. T=850 °C; τ=1.2 s; dp=500-630 µm.

Line 352, 330 ?2∙?−1 (see Table 1), but Table 1 shows proximate, ultimate, ash analysis of treated biomass char and also it it 353 ?2∙?−1 as mentioned in Table 2.

Response:

Agree. The whole section (3.6) was rewritten

Changes to manuscript: [line  307-315]

Both the surface structure and chemical composition of the catalyst influences the catalyst activity. Thus, the impact of the char surface area on naphthalene conversion was studied. The char surface area was measured as explained in section 2.1. Furthermore, the char surface properties were manipulated by producing the pinewood-char at different pyrolysis heating rates (10, 25, and > 50 °C /min) and final pyrolysis temperatures (700, 900, and 950 °C) as shown in Figure 11. It was found the highest naphthalene conversion (99.9 %) was attained at 25 °C /min hearing rate and 700 °C final pyrolysis temperature. However, it was rather difficult to draw conclusions on the impact of surface area at such high naphthalene conversions. On the other hand, these selected conditions were often used at hot gas catalytic cleaning processes.

Figure 11 Impact of pyrolysis conditions on char surface area and naphthalene conversion. τ= 0.25 s; dp= 100-500 µm.

Typo corrections

tars to tar, line 48

Response:

Agree. Although tars and tar are used interchangeably in the literature. We used one term “tar” to represent both and tars was changed to tar in all the manuscript.

Changes to manuscript: corrected in three places

hydrogen atom to hydrogen, line 64

Response:  the sentence was rephrased.   

Changes to manuscript: [line 67-68]

Greensfelder et al. [11] reported that the biomass char converts the tar into free radical by removing the hydrogen

Bartholomew to Bartholomew et al., line 72

Response: Disagree. It is one author thus no need for et. al. 

Changes to manuscript: None

cokes to coke, line 72

Response: Agree, and all such words were changed to coke. 

 Tars are to tar is, line 78

Response: Agree. Although tars and tar are used interchangeably in the literature. We used one term “tar” to represent both and tars was changed to tar in all the manuscript.

Chars to char, line 103

Response: Agree. Changes were done throughout the manuscript.

 Changes to manuscript: line 323

 Reduction to conversion, line 182, 185

Response: Agree. It was changed in all the manuscript from reduction to conversion to unify the meaning.

 Take care of chars, tars, and cokes to char, tar, and coke throughout the manuscript.

Response: Agree. It was changed in all the manuscript

Round 2

Reviewer 2 Report

Details of reaction kinetics, and pathways for naphthalene and char conversion are now included in the revised version. Therefore, I propose to accept this paper in the Catalysts.

Author Response

Response to Reviewer

The authors would like to thank the reviewer for the precious time and important comments. We have carefully addressed all the comments.  

Comment: Details of reaction kinetics, and pathways for naphthalene and char conversion are now included in the revised version. Therefore, I propose to accept this paper in the Catalysts.

Response: We thank the reviewer for accepting (approving) our manuscript with no further comments.

Reviewer 3 Report

I recommend this work for publication in Catalysts. But, Engish corrections must be done thoroughly.

Author Response

Response to Reviewer 2

The authors would like to thank the reviewer for the precious time and important comments. We have carefully addressed all the comments. The corresponding changes and modifications made in the revised paper are summarized below. Our responses are given in a point-by-point manner below each comment in blue color. Changes to the manuscript are shown below the response to each comment in green color.

Comment: I recommend this work for publication in Catalysts. But, Engish corrections must be done thoroughly.

Response: Thank you for your recommendation. We have revised the grammar, typo-mistakes, and others. The modifications can be tracked in the revised version of the manuscript.